# Screening and Assessment of Hypoglycemic Active Peptide from Natural Edible Pigment Phycobiliprotein Based on Molecular Docking, Network Pharmacology, Enzyme Inhibition Assay Analyses, and Cell Experiments

**DOI:** 10.3390/md23080331

**Published:** 2025-08-17

**Authors:** Zhimin Zhu, Yan Zhang, Bingbing He, Limin He, Guihong Fang, Yi Ning, Pengcheng Fu, Jing Liu

**Affiliations:** 1International Collaborative Research Center for the Development and Utilization of Tropical Food for Special Medical Purpose, School of Public Health, Hainan Academy of Medical Sciences, Hainan Medical University, Haikou 571199, China; 23220951350014@hainanu.edu.cn (Z.Z.); 22220951350069@hainanu.edu.cn (Y.Z.); 18789025256@163.com (B.H.); helimin@muhn.edu.cn (L.H.); fgh19801206@163.com (G.F.); ningyi@vip.163.com (Y.N.); 2State Key Laboratory of Marine Resource Utilization in South China Sea, Hainan University, Haikou 570228, China

**Keywords:** phycobiliprotein, hypoglycemic peptide, molecular docking, network pharmacology, α-glucosidase, DPP-IV, IR-HepG2

## Abstract

Phycobiliproteins have gained increasing attention for their antidiabetic potential, yet the specific bioactive peptides and their targets and molecular mechanisms have remained unclear. In this study, four peptides with potential hypoglycemic activity were identified through virtual screening. Network pharmacology was employed to elucidate their hypoglycemic mechanism in the treatment of T2DM. A subsequent in vitro assay confirmed that the synthesized peptides, GR-5, SA-6, VF-6, and IR-7, exhibited significant inhibitory activity against α-glucosidase and DPP-IV. In insulin-resistant HepG2 models, all four peptides exhibited no cytotoxicity. Among them, GR-5 demonstrated the most promising therapeutic potential by remarkably enhancing cellular glucose consumption capacity. Furthermore, GR-5 administration substantially increased glycogen synthesis and enzymatic activities of hexokinase and pyruvate kinase with statistically significant improvements compared to the control groups. This study provides novel peptide candidates for T2DM treatment and validates an integrative strategy for targeted bioactive peptide discovery, advancing the development of algal protein-based therapeutics.

## 1. Introduction

Type 2 diabetes mellitus (T2DM) is a common chronic metabolic disorder, ranking just behind cancer and cerebrovascular diseases in terms of incidence rate [1]. Research has demonstrated a strong link between T2DM and chronic inflammation, particularly within metabolically active organs such as the liver, kidneys, heart, and pancreas. In these tissues, macrophages play a crucial role in regulating insulin homeostasis [2]. Insulin, a hormone produced by the pancreas, facilitates the conversion of blood sugar into energy to maintain normal physiological functions. When blood sugar levels rise, the body compensates by increasing insulin secretion to restore balance, a process that may lead to the onset of insulin resistance [3]. One of the primary strategies in the T2DM treatment involves the inhibition of key enzymes associated with glucose metabolism. In particular, dipeptidyl peptidase-IV (DPP-IV) inhibitors and α-glucosidase inhibitors are commonly employed to manage postprandial blood glucose levels [4].

α-Glucosidase is a key intestinal enzyme involved the terminal digestion of carbohydrates and plays a crucial role in regulating postprandial blood glucose levels [5]. Carbohydrate digestion proceeds through three phases: initial luminal hydrolysis, followed by α-amylase-catalyzed disaccharide production, and final monosaccharide release via stereoselective cleavage by α-glucosidase at the brush border membrane [6]. This multicomponent enzymatic cascade inherently positions α-glucosidase inhibition as a therapeutic strategy for improving glycemic control [7]. α-Amylase, a hydrolase enzyme predominantly synthesized in the pancreatic and salivary gland systems [8], catalyzes the hydrolysis of α-1,4-glucosidic bonds in starch molecules, breaking them down into maltose and glucose subunits [9]. The inhibition of α-amylase activity effectively reduces the degradation of polysaccharide, thereby mitigating postprandial hyperglycemia by delayed carbohydrate absorption [10]. DPP-IV is a dual-form serine protease existing in membrane-anchored and soluble plasma forms [11]. It specifically cleaves alanyl-proline bonds in bioactive peptides, degrading key regulators such as glucagon-like peptide-1 (GLP-1) and glucose-dependent insulinotropic polypeptide [12]. The cleavage of GLP-1 by DPP-IV promotes glucagon secretion while suppressing insulin production, thereby contributing to elevated blood glucose concentrations [13]. Therapeutic DPP-IV inhibitors function as competitive molecular antagonists targeting the catalytic pocket, effectively preserving endogenous GLP-1 bioavailability without compromising the enzyme’s structural integrity [14].

Bioactive peptides, composed of 2 to 100 amino acids, perform a wide range of physiological functions, including the regulation of growth, immunity, and metabolism. Owing to their high specificity and structural diversity, bioactive peptides have shown considerable promise in drug development and therapeutic applications. In the treatment of T2DM, several bioactive peptides have demonstrated significant inhibitory effects. For instance, Xiao et al. used a combination of computational tools and in vitro methods to identify a potential endogenous antidiabetic peptide, PVTQPL, from human milk [15]. Chandrasekaran et al. isolated a peptide, SPGAGKG, from sprouted chickpea that exhibited strong DPP-IV inhibitory effect [16]. Additionally, Li extracted and characterized a peptide, VGPDGSPDPL, from seaweed with notable α-glucosidase inhibition [17].

In recent years, network pharmacology and molecular docking have become valuable tools for identifying novel drug targets and elucidating the molecular mechanisms underlying therapeutic actions. For example, Liu et al. used network pharmacology to identify 250 core targets for the ginsenosides Rk1 and Rg5 in T2DM treatment. Enrichment analysis using the Kyoto Encyclopedia of Genes and Genomes (KEGG) revealed their involvement in insulin resistance pathways, and the molecular docking indicated strong Rk1 and Rg5 binding affinities to Akt1 [18]. Xu et al. identified 34 key therapeutic targets for T2DM associated with nonalcoholic fatty liver disease (NAFLD), including IL6, IL1B, VEGFA, PTGS2, and CCL2. Gene Ontology (GO) and KEGG pathway enrichment analysis suggested that 34 compounds from Ginkgo biloba may exert their therapeutic effects by modulating responses to bacterial-derived molecules, lipids, and hormones. Additionally, molecular docking further confirmed strong binding interactions between active Ginkgo compounds and these targets [19]. Therefore, the integration of network pharmacology and molecular docking offers a rapid and efficient approach for identifying antidiabetic compounds in complex natural product systems. This strategy not only accelerates drug discovery but also provides a holistic understanding of underlying mechanisms [20].

*Spirulina*, known for its exceptionally high protein content under artificial culture conditions, is widely used as a health food and dietary supplement. Phycobiliproteins are water-soluble pigment–protein complexes which are known for their antioxidant, anti-inflammatory, and immunomodulatory properties [21]. Li et al. have also highlighted their antidiabetic potential by using either biochemical or cellular techniques [22]. In our preliminary investigation, various proteases sourced from distinct origins including pepsin, trypsin, alkaline protease, papain, and bromelain were used to facilitate the enzymatic hydrolysis of phycobiliproteins. Through LC-MS/MS analysis, we successfully identified 1333 unique peptide sequences derived from phycobiliproteins. Of these, 73 peptides were predicted to have high bioactivity based on PeptideRanker scores. Subsequent in vivo studies demonstrated that the active peptides significantly mitigated the T2DM symptoms. However, the specific hypoglycemic peptides responsible for this effect have not yet been isolated, and the molecular mechanisms by which these hypoglycemic peptides reduce the expression of target proteins to lower blood glucose levels remain insufficiently understood, thus limiting their potential as functional food ingredients.

To address this gap, the present study proposes an integrative computational–experimental framework for the discovery of hypoglycemic peptides from *Spirulina* phycocyanin, comprising three systematically coordinated phases: (1) structure-based virtual screening via molecular docking was conducted to prioritize peptides with multi-enzyme binding potential; (2) network pharmacology was employed to identify key therapeutic targets, followed by mechanistic elucidation through GO/KEGG enrichment analysis; (3) selected peptides were synthesized using solid-phase peptide synthesis, and their bioactivities were validated through in vitro triple-enzyme inhibition assays targeting α-glucosidase, α-amylase, and DPP-IV. Additionally, their hypoglycemic potential was confirmed in insulin-resistant HepG2 (IR-HepG2) models. This multidimensional strategy not only unveils the polypharmacological mechanisms of peptide-based glucose regulation but also provides technical references for developing marine-derived antidiabetic therapeutics.

## 2. Results and Discussion

### 2.1. Molecular Docking Analysis

To elucidate the molecular interactions between candidate peptides and target proteins, molecular docking simulations were conducted to quantitatively assess binding potential, expressed as binding energy values (kcal/mol) [23] in Appendix A. The computational analysis revealed four lead peptide candidates, VF-6 (Val-Ala-Phe-Gly-Arg-Phe), GR-5 (Gly-Tyr-Tyr-Leu-Arg), SA-6 (Ser-Tyr-Phe-Asp-Arg-Ala), and IR-7 (Ile-Ala-Ser-Tyr-Phe-Asp-Arg), exhibiting superior binding affinities, each demonstrating binding energies lower than −8.0 kcal/mol (Table 1). Previous studies have highlighted the critical role of hydrogen bond interactions in stabilizing docked complexes and facilitating enzymatic–substrate interactions [24]. Using PyMOL visualization software, we analyzed hydrogen bonding patterns between peptides and target proteins, recording both the number of hydrogen bonds and the shortest hydrogen bond distances, as depicted in Table 1. These data underscore the high binding capacity of the selected peptide. Notably, peptide VF-6 exhibited the lowest binding energy of −9.4 kcal/mol with DPP-IV forming up to 13 hydrogen bonds with receptor residues, indicating a strong interaction potential with the proteins.

The DPP-IV active site comprises three functional sub-regions: the catalytic triad (Ser630, Asp708, His740); the oxyanion hole (Tyr547, Ser631); and a salt-bridge region (Glu205, Glu206, Tyr662). Additional residues such as Arg125, Ser209, Phe357, Arg358, Tyr547, Ser631, Val656, Trp659, Tyr662, Tyr666, Asn710, and Val711 occupy substrate-binding pockets S1 and S2 [25]. As illustrated in Figure 1A, VF-6 interacts with DPP-IV via hydrogen bonds with Arg125, Glu206, Arg358, Val546, Tyr547, Lys554, Trp629, and Ser630; hydrophobic contacts with Phe357, Val656, Tyr666, and Val711; π-π stacking with Trp629, Tyr662, and Tyr666; and salt bridges’ formation with Arg125, Asp545, and His740. As depicted in Figure 1B, VF-6 engages with α-glucosidase through hydrogen bonds with the following: Glu377, Asp333, and Asn301, along with extensive hydrophobic interactions with nearby residues. The docking was targeted to the active site of α-glucosidase, specifically centered on residues Tyr65, Ile146, Phe166, Arg200, Thr226, Gly228, Ala229, Glu271, Phe297, Asn301, His332, Asp333, and Arg400, which are known to be critical for catalytic activity. The binding pocket was defined based on previous structural studies [26].

Given their capacity to form multiple stabilizing hydrogen bonds and their low binding energies, with both DPP-IV (4a5s) and α-glucosidase (3wy1), these four peptides were selected for future experimental validation.

### 2.2. Hypoglycemic Mechanism Analysis of Active Peptides in T2DM Treatment

#### 2.2.1. Screening for Potential T2DM-Related Interaction Targets

Using the Swiss Target Prediction platform, a total of 216 potential targets were identified for the active peptides. Additionally, 1908 T2DM-related targets were retrieved from the GeneCards database and 519 targets from the OMIM-GENE-MAP database. After merging and de-duplicating these datasets, a total of 2372 unique T2DM-related targets were compiled. Cross-referencing this list with peptide-associated targets revealed 83 overlapping genes, as illustrated in Figure 2. Each peptide was associated with multiple T2DM-related targets: VF-6 (48), GR-5 (49), SA-6 (41), and IR-7 (50).

To visualize these relationships, a “component–target” network was constructed using Cytoscape 3.9.1 (Figure 3). The network illustrates the multitarget potential of several peptides. For example, the component VF-6 interacts with 42 targets, including NTSR2, PRSS1, and F2, whereas SA-6 is associated with 32 targets, such as IL1B, AKT1, and MAPK1. Furthermore, several targets are shared among multiple peptides, such as CASP3 target, which binds to FDAFTK, IR-7, and RPDVVSP, among others. These findings suggest the possibility of synergistic therapeutic effects, whereby multiple peptides may cooperatively modulate several molecular targets, thereby enhancing their overall efficacy in regulating glucose metabolism and mitigating the progression of T2DM.

#### 2.2.2. PPI Network Analysis

To identify key molecular targets and better understand protein–protein interaction relationships, a PPI network was constructed and analyzed as shown in Figure 4. In this network, nodes represent protein targets, and edges represent interactions between them. The resulting network consisted of 82 nodes and 702 edges reflecting a dense web of interactions.

Topological analysis of the PPI network was performed using the Cytoscape plug in, CentiScape. Based on predefined thresholds (Degree = 17.12; Betweenness Centrality = 75.32; Closeness Centrality = 0.007), a total of 19 core targets were identified as hub proteins. The network topology parameters for these core targets are summarized in Table 2. The resulting 19 core targets are a subset of the 83 shared targets due to their high degree of interaction connectivity. This refinement allows us to focus on the most functionally important proteins for additional enrichment analysis.

Among the hub nodes, AKT1 exhibited the highest degree value, underscoring its central role in regulating various biological processes, including metabolism, cell proliferation, survival, growth, and angiogenesis. AKT1 is a serine/threonine kinase family and a key component of the phosphatidylinositol 3-kinase/protein kinase B (PI3K/AKT) signaling cascade [27]. Activation of this pathway results in the recruitment and phosphorylation of AKT1 on the cell membrane which in turn activates downstream targets such as glucose transporters, thereby regulating glucose uptake and homeostasis [28]. In addition to its role in enhancing insulin sensitivity, AKT1 is critical for maintaining pancreatic β cell function, survival, and insulin secretion as key factors in the management of T2DM [29]. Nevertheless, the accumulation of advanced glycation end products (AGEs) can impair β-cell function by inducing apoptosis, ultimately leading to inadequate insulin secretion [30].

Another important target, Interleukin-1β (IL-1β) is an essential pro-inflammatory cytokine involved in initiating antimicrobial immune responses. In the context of autoimmune diseases, IL-1β activity is aberrantly heightened, prompting the utilization of IL-1β inhibitors as therapeutic agents for certain autoimmune conditions. IL-1β functions as a detector, directly recognizing pathogen-associated molecular patterns through a parallel mechanism with the host’s inflammatory complex [31]. The literature suggests that IL1B also contributes to renal fibrosis in diabetic nephropathy, further establishing its relevance in T2DM pathogenesis [32].

Src, a non-receptor tyrosine kinase, regulates a range of cellular functions; its aberrant activation has been linked to impaired insulin signaling and β cell dysfunction [33]. Similarly, signal transducer and activator of transcription 3 (STAT3) plays a pivotal role in signal transduction from the plasma membrane to the nucleus and mitochondria, where it regulates gene transcription and mitochondrial function, influencing metabolism, immunity, and inflammation [34]. Cysteine aspartase 3 (CASP3) is a key executioner enzyme in apoptosis. Suppressing CASP3 activity may reduce pancreatic β-cell death, thus preserving insulin secretion and improving insulin resistance [35]. Matrix metalloprotein 9 (MMP9), a member of the matrix metalloprotein family, predominantly regulates the synthesis and secretion of neutrophils and macrophages, engaging in various physiological and pathological processes such as angiogenesis and inflammatory response, and can exacerbate inflammation by disrupting tissue. Its expression is elevated in T2DM patients with microvascular complications and may promote pathological angiogenesis and immune cell infiltration [36].

#### 2.2.3. GO and KEGG Pathway Enrichment Analysis

To further explore the functional roles of the 83 common targets identified, GO and KEGG pathway enrichment analyses were performed using the Metascape database, applying a significant level of *p* < 0.01. GO enrichment analysis resulted in a total of 1216 terms, comprising 1061 biological processes (87.25%), 93 molecular functions (7.65%), and 62 cellular components (5.10%). The top 10 terms from each category were selected for graphical visualization in the form of a bar chart (Figure 5). Key biological processes included cellular responses to peptides, hormones, and nitrogen-containing compounds. Regarding molecular functions, enriched terms focused on endopeptidase activity, peptidase activity, and serine-type peptidase activity. Enriched cellular components were primarily associated with extracellular matrix, external encapsulating structures, and regions of the extracellular matrix region.

KEGG pathway analysis identified 162 signaling pathways, with the top 10 pathways most relevant to T2DM, visualized in a bubble chart (Figure 6). In this chart, the horizontal axis represents the gene ratio, and the vertical axis lists the enriched pathways. The color gradient corresponds to *p*-values, and bubble size indicates the number of involved genes. Notably enriched pathways included the PI3K-Akt signaling pathway, insulin signaling pathway, HIF-1 signaling, and inflammatory response pathways, all of which are closely linked to T2DM pathogenesis.

### 2.3. Validation of Synthetic Active Peptides

#### 2.3.1. Purity and Molecular Weight of the Hypoglycemic Active Peptides

The four lead peptides VF-6, GR-5, SA-6, and IR-7, were synthesized using solid-phase peptide synthesis (SPPS). The resulting peptides demonstrated high purity, with measured values of 98.026%, 98.259%, 98.522%, and 98.187%, respectively. Corresponding molecular weights were confirmed as 695.81 Da, 670.76 Da, 757.78 Da, and 870.94 Da, respectively.

#### 2.3.2. α-Glucosidase Inhibition Activity

As shown in Figure 7A, the peptide GR-5 exhibited a 3.2-fold greater inhibitory effect on α-glucosidase activity compared to the clinical drug acarbose, with respective a IC_50_ value of 0.1148 ± 0.0084 versus 0.3639 ± 0.0111 mg/mL. This enhanced inhibitory potency correlated with improved glycemic modulation capacity at submilligram concentrations, maintaining linear dose–response characteristics below 1.0 mg/mL.

Conversely, VF-6’s exhibited limited inhibition, with a higher IC_50_ of 0.7824 ± 0.0376 mg/mL, indicating weaker enzyme interaction. IR-7 displayed a comparable profile to VF-6 at concentrations above 0.5 mg/mL. SA-6 showed the weakest inhibition in the group, reflecting minimal efficacy in this context.

When compared to previous research, our results mark a significant development in α-glucosidase inhibition. Meanwhile, Flores-Medellín’s fermented legume hydrolysates achieved IC_50_ = 5.55 mg/mL [37], and Ye’s cereal phenolics achieved 0.20 mg/mL [38]. The GR-5 peptide, on the other hand, demonstrated noticeably higher inhibitory efficacy, with an IC_50_ of 0.1148 mg/mL. More significantly, its potency is equivalent to that of the pharmaceutical α-glucosidase inhibitor acarbose, which has been shown to have IC_50_ values between 0.195 and 0.358 mg/mL in comparable in vitro settings [39,40,41]. In addition to outperforming the majority of natural plant-based inhibitors by a factor of over 27.7, this comparison shows that GR-5 reaches inhibition levels that are comparable to those of clinically approved medications, indicating its potential as a next-generation antidiabetic contender.

#### 2.3.3. α-Amylase Inhibition Activity

Several natural bioactive compounds have demonstrated potent α-amylase inhibitory effects in vitro. Notably, Saidi et al. documented significant anti-α-amylase activity in EtOAc and n-BuOH extracts, with IC_50_ values of 52.5 ± 1.2 µg/mL and 36.3 ± 1.1 µg/mL, respectively [42]. In a related study, Ge identified two novel peptide inhibitors (FPSPPLVA and GPPMPPPPLP) with IC_50_ values of 0.92 ± 0.27 mg/mL and 2.02 ± 0.2 mg/mL [43]. Acarbose exhibits IC_50_ values against α-amylase in the range of 0.258–0.50 mg/mL under similar in vitro conditions [44,45].

In this study, comparative analysis of the four synthesized peptides demonstrated the following potency hierarchy against α-amylase (Figure 7B):

SA-6 (IC_50_ = 0.8603 ± 0.1312 mg/mL) > VF-6 (IC_50_ = 0.8297 ± 0.0987 mg/mL) > acarbose (IC_50_ = 0.5195 ± 0.0714 mg/mL) > GR-5 (IC_50_ = 0.308 ± 0.0604 mg/mL) > IR-7 (IC_50_ = 0.1292 ± 0.0523 mg/mL).

Interestingly, IR-7 exhibited superior inhibitory efficacy at concentrations below 1 mg/mL but displayed a paradoxical decline in activity beyond this threshold. At 2 mg/mL, its inhibitory performance demonstrated inferiority relative to both the control group and other peptides, which may indicate enzyme saturation at 1 mg/mL, where all available binding sites are occupied, thus preventing further enhancement of inhibition at higher concentrations.

#### 2.3.4. DPP-IV Inhibition Activity

Comparative evaluation of four peptides for DPP-IV inhibition, as seen in Figure 7C, revealed the following IC_50_ values:

VF-6 (IC_50_ = 0.9468 ± 0.1933 mg/mL);

IR-7 (IC_50_ = 0.9089 ± 0.1672 mg/mL);

GR-5 (IC_50_ = 4.677 ± 0.935 mg/mL);

SA-6 (IC_50_ = 8.344 ± 2.853 mg/mL).

VF-6 and IR-7 emerged as potent DPP-IV inhibitors, exhibiting concentration-dependent efficacy. At a concentration of 10 mg/mL, IR-7 exhibited 32% greater inhibition than sitagliptin, a clinically approved DPP-IV inhibitor. In contrast, GR-5 and SA-6 showed weaker inhibition, with IC_50_ values much higher than those of VF-6 and IR-7. In alignment with these results, Bollati et al. reported IC_50_ values of 1.15 mg/mL and 1.33 mg/mL for soybean and pea protein-derived hydrolysates, respectively [46]. For reference, Sitagliptin, a clinically approved DPP-IV inhibitor, has been reported to exhibit IC50 values of approximately 18 nM under similar in vitro conditions [47].

These findings underscore the potent antihyperglycemic potential of *Spirulina*-derived phycobiliprotein peptides. Pharmacological characterization revealed distinct target selectivity among the peptides: VF-6, GR-5, SA-6, and IR-7 demonstrated differential inhibition patterns against carbohydrate-digestive enzymes and DPP-IV. All peptides exhibited broad-spectrum α-amylase suppression, while GR-5 showed superior α-glucosidase inhibition. Collectively, these peptides demonstrated multienzyme modulation and distinct target selectivity, with VF-6 and IR-7 emerging as prioritized candidates for the development of next-generation antidiabetic agents targeting DPP-IV pathways.

### 2.4. Effect of Peptides on IR-HepG2 Cells

#### 2.4.1. Establishment of the IR-HepG2 Cell Model

Insulin resistance, a hallmark of T2DM, was modeled in HepG2 cells through chronic hyperinsulinemia exposure to evaluate the therapeutic potential of the peptides [48]. The establishment of the IR-HepG2 cell model and cellular viability assessment were implemented as the experimental workflow in Figure 8A,B. Treatment with 10^−4^ mol/L insulin resulted in a marked reduction in HepG2 cell viability to 38.45%, demonstrating significant inhibition compared to the normal control (*p* < 0.0001). Furthermore, exposure to 2.5 mmol/L concentrations of peptides VF-6, SA-6, and IR-7 significantly suppressed cell viability relative to the blank control (*p* < 0.0001). Notably, insulin concentrations ranging from 10^−5^ to 10^−9^ mol/L and other tested peptide concentrations, including GR-5, did not induce cytotoxicity, confirming the biocompatibility of the peptides.

As shown in Figure 9, HepG2 cells were exposed to 10^−7^ mol/L insulin for 36 h to establish the IR-HepG2 cell model. The successful induction of insulin resistance was confirmed by a significant decrease in glucose consumption, a key indicator of impaired insulin signaling. To further validate the stability of the insulin resistance model, we optimized the insulin exposure parameters. After 24 h of exposure, the IR-HepG2 cells exhibited the most significant decrease in glucose consumption compared to normal control, confirming that 24 h is the optimal treatment duration for subsequent antidiabetic peptide evaluations.

#### 2.4.2. Effect of Peptides on Glucose Consumption in IR-HepG2 Cells

Impaired glucose uptake in insulin-resistant cells contributes to elevated blood glucose levels and the progression of T2DM. Glucose consumption was measured as a direct indicator of glucose metabolism in the IR-HepG2 cellular model to evaluate the effects of bioactive peptides on glucose utilization [49].

As shown in Figure 10, the model control group exhibited a significant reduction in glucose consumption compared to the normal group, showing a 58.9% decrease (*p* < 0.0001). Treatment with metformin in the positive control group exhibited a significant increase in glucose consumption, especially at a concentration of 2 mmol/L, where glucose consumption increased by 105% compared to baseline. This confirmed the successful establishment of the IR-HepG2 model.

At concentrations of 1 and 2 mmol/L, all four peptides significantly increased glucose consumption compared to the model group (*p* < 0.01). GR-5 significantly increased glucose consumption by 41.4% and 60.8% at 1 mmol/L and 2 mmol/L treatments, respectively. SA-6 elevated glucose uptake with 13.9% and 67.3% enhancements under equivalent doses. VF-6 demonstrated dose-dependent effects on glucose utilization, showing 47.0% and 79.2% increases at 1 mmol/L and 2 mmol/L concentrations. Notably, IR-7 exhibited paradoxical responses: a 75.0% elevation at 1 mmol/L contrasted with a reduced 54.9% increase at 2 mmol/L treatment.

Cytotoxicity assays confirmed that all peptides were non-toxic at these concentrations. Based on these results, GR-5, SA-6, and VF-6 were selected for further experiments at 2 mmol/L, while IR-7 was used at 1 mmol/L.

#### 2.4.3. Effect of Peptides on Glucose Metabolism in IR-HepG2 Cells

Glycogen synthesis and glycolysis are key metabolic pathways regulating glucose utilization and maintaining energy homeostasis [50]. Hexokinase (HK) and pyruvate kinase (PK) are rate-limiting enzymes at the initial and terminal steps of glycolysis, respectively, and their activities directly influence glucose metabolic efficiency. Impaired hepatic glycogen synthesis and suppressed HK/PK activities are critical factors for glucose metabolic disorders [51]. This study systematically evaluated the regulatory effects of four hypoglycemic peptides (GR-5, SA-6, VF-6, and IR-7) on glycogen synthesis and key glycolytic enzyme activities using the IR-HepG2 cell model.

As shown in Figure 11A, the control group exhibited a 44.5% reduction in glycogen content compared to the normal group (*p* < 0.001). Metformin and GR-5 interventions significantly increased glycogen levels by 24.2% and 24.0%, respectively (*p* < 0.05). SA-6, VF-6, and IR-7 treatments increased glycogen content by 14.3%, 6.9%, and 9.4%, respectively, demonstrating differential improvements compared to baseline levels.

Enzymatic assays (Figure 11B,C) revealed that HK and PK activities in the model control group decreased by 58.8% and 24.4%, respectively, compared to the normal group (*p* < 0.01). Metformin treatment markedly reversed enzyme inhibition, elevating HK and PK activities by 54.9% and 21.1%, respectively. Following hypoglycemic peptide interventions, HK activity increased by 32.7%, 19.5%, 12.4%, and 5.3% across sequential treatments, while PK activity rose 17.5%, 9.4%, 2.7%, and 0.9%. GR-5 showed the most significant activation of both enzymes compared to the model control group (*p* < 0.05).

The four hypoglycemic peptides alleviated glucose metabolic dysregulation in IR-HepG2 cells by enhancing glycogen synthesis and restoring HK/PK activities, with GR-5 demonstrating the strongest efficacy. These findings confirm their protective effects against insulin resistance-associated metabolic disorders, highlighting their potential as functional hypoglycemic agents or therapeutic candidates for the treatment of T2DM.

## 3. Materials and Methods

### 3.1. Materials and Reagents

The four peptides were synthesized by Shanghai Science Peptide Biological Technology Co., Ltd. (Shanghai, China). α-Glucosidase (50 U/mg), insulin, and metformin were purchased from Shanghai Yuanye Bio-Technology Co., Ltd. (Shanghai, China). α-Amylase (50 U/mg), 4-nitrophenyl α-D-glucopyranoside (pNPG) (CAS: 3767-28-0), soluble starch, 3,5-dinitrosalicylic acid, and potassium sodium tartrate tetrahydrate solution were purchased from Shanghai Macklin Biochemical Co., Ltd. (Shanghai, China). DPP-IV Inhibitor Screening Kit (E-BC-D007) was purchased from Elabscience Biotechnology Co., Ltd. (Wuhan, China). Acarbose hydrates were purchased from Aladdin. HepG2 cells and the Cell Counting Kit 8 (CCK-8) were purchased from Xiamen Immocell Biotechnology Co., Ltd. (Xiamen, China). Glucose assay kits and Pyruvate Kinase (PK) activity assay kits were obtained from Abbkine Scientific Co., Ltd. (Wuhan, China). Hexokinase (HK) activity assay kits were procured from Nanjing Jiancheng Bioengineering Institute (Nanjing, China). Glycogen assay kits were sourced from Beijing Solarbio Science & Technology Co., Ltd. (Beijing, China). All other chemicals and reagents used in this study were of analytical grade.

### 3.2. Bioactive Peptides from Phycobiliproteins

Peptide sequences were identified from hydrolysates of natural edible pigment phycobiliproteins. Preliminary activity evaluation was conducted using Peptide Ranker database (http://distilldeep.ucd.ie/PeptideRanker/ (accessed on 21 January 2024)) (set Score > 0.5), as reported in our previous studies [52]. The identified peptides sequences are listed in Appendix A.

### 3.3. Virtual Screening of Hypoglycemic Active Peptides by Molecular Docking

Molecular docking was performed using open-source software AutoDock Vina 1.1.2. Although AutoDock is traditionally optimized for small molecule docking, it has been successfully applied in several studies for short peptide–protein interactions due to its accessible and flexible scoring functions [53]. Peptide structures were initially drawn using ChemDraw 21.0.0 and optimized via Chem3D 21.0.0. Molecular docking was performed after minimizing the energy of the peptide structure. In this study, α-glucosidase (PDB ID: 3wy1) and DPP-IV (PDB ID: 4a5s) were selected as target receptor proteins. Their X-ray crystal structure was downloaded from the Protein Data Bank (PDB). Pre-docking processing steps, including dehydration, hydrogenation, and charge balance of both receptor protein and ligand were completed using AutoDock Tools (MGLTools-1.5.6). The docking box dimensions and active site coordinates were set as follows: 20Å × 26Å × 24Å (3wy1) and 32Å × 26Å × 28Å (4a5s), the coordinates of the active center were (2.473, −18.215, 10.254) and (13.984, 29.803, 56.514), respectively. The docking results were analyzed and visualized by PyMOL 2.5.5 [54]. The databases and platform websites in this procedure are listed in Appendix A.

### 3.4. Assessment of Regulatory Mechanisms Through Network Pharmacology

Our previous studies revealed that phycobiliproteins’ hydrolysates can regulate glucose metabolism [55]. In this study, we applied network pharmacology to explore the potential hypoglycemic pathways of the active peptides.

Target prediction for the screened peptides was conducted using the Swiss Target Prediction platform, with the species parameter set to *Homo sapiens* and all other settings kept at the default. To identify disease-related targets, “Type 2 diabetes mellitus” was used as the search keyword in both GeneCards and OMIM-GENE-MAP databases. Targets from these two databases were combined to form a comprehensive dataset of T2DM-associated genes. The intersecting targets between the peptide-predicted targets and T2DM-related genes were identified using bioinformatics tools.

To delve deeper into the molecular interaction of active peptides on T2DM, the overlapping targets were uploaded to the STRING database for protein–protein interaction (PPI) network analysis, with species set to *Homo sapiens* and a minimum interaction score threshold of 0.4 (medium confidence). The resulting data were saved in “tsv” format into Cytoscape 3.9.1. Using the Centiscape plugin, we performed topological analysis to identify key regulatory hub genes.

For functional annotation, GO and KEGG enrichment analysis were conducted using the Metascape platform. Both the input species and analysis species were set to *H. sapiens* with significant threshold defined as *p*-value < 0.01. Visualization of enrichment results was completed using https://www.bioinformatics.com.cn (accessed on 11 March 2024), an online platform for data analysis and visualization.

### 3.5. Solid-Phases Synthesis and Validation of Hypoglycemic Active Peptides

The hypoglycemic active peptides were synthesized via solid-phase synthesis by Shanghai science peptide Co., Ltd. (Shanghai, China). The purity and molecular weight of the synthesized peptides were confirmed using HPLC-MS. The α-glucosidase inhibitory activity was assessed a modified version of previously reported methods [56] by adding 100 μL of p-NPG (1 mM) after 50 μL of peptide solution (0.125–2.0 mg/mL), and 50 μL of α-glucosidase (1 u/mL) were incubated at 37 °C for 15 min. Absorbance was measured at 405 nm after the reaction was stopped by adding 100 μL of NaCO_3_ after another 30 min of incubation. Similarly, α-amylase inhibitory activity was determined based on established protocols [53] with minor modifications. Briefly, 100 μL of peptide solution (0.125–2.0 mg/mL) was mixed with 100 μL of α-amylase (1 U/mL) and incubated at 37 °C for 10 min. Then, 200 μL of pretreated soluble starch solution (2%, denatured at 95 °C for 8 min) was added, and the mixture was further incubated at 37 °C for 20 min. The reaction was stopped by adding 200 μL of DNS reagent (1% DNS; 12% sodium potassium tartrate in 0.4 M Na_2_CO_3_), followed by boiling at 100 °C for 15 min. After cooling, 200 μL of each reaction mixture was transferred to a 96-well plate, and absorbance was measured at 540 nm. DPP-IV inhibitory activity was evaluated using a DPP-IV inhibitor screening kit (Elabscience) following the manufacturer’s instructions with slight modifications. Peptide solutions were prepared at gradient concentrations (0.125, 0.25, 0.5, 1, 1.5, 2, 4, and 10 mg/mL). In a 96-well plate, 20 μL of enzyme solution and 30 μL of peptide sample were added to each well, mixed briefly, and incubated at 37 °C for 10 min. Then, 170 μL of reaction working solution was added, followed by a second incubation at 37 °C for 30 min. Fluorescence intensity was measured at an excitation wavelength of 360 nm and an emission wavelength of 460 nm. The IC_50_ values were determined by fitting dose–response curves using nonlinear regression analysis with a variable slope in GraphPad Prism 9. All IC_50_ results are expressed as mean ± SD based on three independent experiments.

### 3.6. Cell Culture and Model Establishment

HepG2 cells were maintained in Dulbecco’s Modified Eagle Medium (DMEM) supplemented with 10% fetal bovine serum and 1% penicillin–streptomycin under controlled conditions at 37 °C in a humidified atmosphere containing 5% CO_2_ for 24 h. To induce insulin resistance, cells were treated with insulin at concentrations of 0, 10^−4^, 10^−5^, 10^−6^, 10^−7^, 10^−8^, and 10^−9^ μmol/L for varying exposure periods (12, 24, 36, or 48 h), followed by a post-treatment incubation phase of equal length under standard culture conditions, according to the protocol described by [57].

Cell viability was assessed using the Cell Counting Kit 8 (CCK-8) assay. Glucose consumption was measured via a glucose detection kit. The optimal insulin concentration for establishing the insulin-resistant HepG2 cell models was determined based on absence of cytotoxicity and significantly reduced glucose consumption.

### 3.7. Determination of Glucose Metabolism

The IR-HepG2 cellular model was established as described in Section 3.6. The experimental design included the following groups: a normal (untreated) control group, an insulin resistance (IR)-induced model group, a positive control group treated with metformin, and peptide intervention groups treated with peptide solution. Cell viability was evaluated in all groups to determine appropriate peptide concentrations and incubation times based on the criteria of non-cytotoxicity and maximal glucose consumption.

Glucose consumption was measured using a commercial glucose assay kit according to the manufacturer’s protocol. Intracellular glycogen content was quantified with a glycogen detection kit, while hexokinase (HK) and pyruvate kinase (PK) enzyme activities were determined using specific assay kits, following the provided instructions.

### 3.8. Statistical Analysis

All experiments were independently repeated at least three times. Data are presented as mean ± standard deviation (SD). Statistical differences among groups were analyzed using one-way analysis of variance (ANOVA), followed by Duncan’s multiple range test using SPSS 26.0 (IBM Corp., Armonk, NY, USA). A *p* value of <0.05 was considered to be statistically significant. Data visualization was performed using GraphPad Prism 9.0.0 and Origin 2021b (OriginLab Inc., Northampton, MA, USA).

## 4. Conclusions

This study utilized a synergistic approach combining network pharmacology and molecular docking methodologies to identify potential antihyperglycemic peptides derived from the hycobiliproteins of *Spirulina*. The research highlights the key targets and mechanisms by which these active peptides modulate blood glucose levels. Among the proteolytic peptide fragments, we identified several peptides with promising antihyperglycemic properties, forming an intricate network that facilitates glucose reduction. These peptides exhibit characteristics indicative of multi-target and multi-pathway hypoglycemic effects. Through PPI network analysis, we pinpointed 19 core targets, including AKT1, IL1B, Src, STAT3, CASP3, and MMP9, among others. These targets are implicated in biological processes such as apoptosis, glycometabolism, lipid metabolism, and inflammatory response; collectively they contribute to the observed hypoglycemic effect.

Our experimental investigations demonstrated that four bioactive peptides (GR-5, SA-6, VF-6, and IR-7) exhibited dose-dependent inhibitory effects on three critical diabetes-related enzymes: α-glucosidase, α-amylase, and DPP-IV. Notably, Although VF-6 demonstrated more favorable docking scores, GR-5 exhibited superior in vitro inhibitory activity against both α-glucosidase and α-amylase. This discrepancy may result from limitations in the docking model, which does not fully account for dynamic binding and peptide stability under physiological conditions. It should be noted that AutoDock has limitations even though it has been used successfully in a number of studies involving short peptides. More sophisticated and precise peptide–protein docking tools, like HADDOCK, Rosetta FlexPepDock, or CABS-dock, which provide improved modeling of peptide backbone flexibility, solvent effects, and dynamic binding, should be used in future research. By incorporating these tools, binding mode predictions may become more accurate and more accurately represent the biological interactions seen in vitro.

GR-5 emerged as the most promising candidate in cellular models, significantly enhancing glucose consumption capacity, promoting glycogen synthesis, and upregulating key glycolytic enzymes HK and PK in IR-HepG2 cells. These cellular improvements suggest that GR-5’s operates through a dual mechanism, involving both enzymatic inhibition and metabolic pathway modulation. However, the extrapolation of these findings requires caution, as the peptides’ in vivo efficacy, pharmacokinetic profile, and underlying mechanisms still need to be fully explored through systematic animal studies.

In summary, this study presents a strategy that combines network pharmacology prediction with molecular docking validation to systematically identify antihyperglycemic peptides derived from *spirulina* phycocyanin. Through enzyme inhibition assays, cellular validation, and computational simulations, we not only confirmed the hypoglycemic bioactivity of these peptides, but also uncovered their potential mechanisms, which involve the dual regulation of carbohydrate-digesting enzymes and insulin signaling pathways. These findings establish a three-dimensional mechanistic framework connecting in silico predictions with in vitro validations, offering theoretical insights the hypoglycemic action phycocyanin and providing technical references for development of algae-based nutraceuticals. Future research should prioritize in vivo verification and clinical translation of these bioactive peptides.

## Figures and Tables

**Figure 1 marinedrugs-23-00331-f001:**
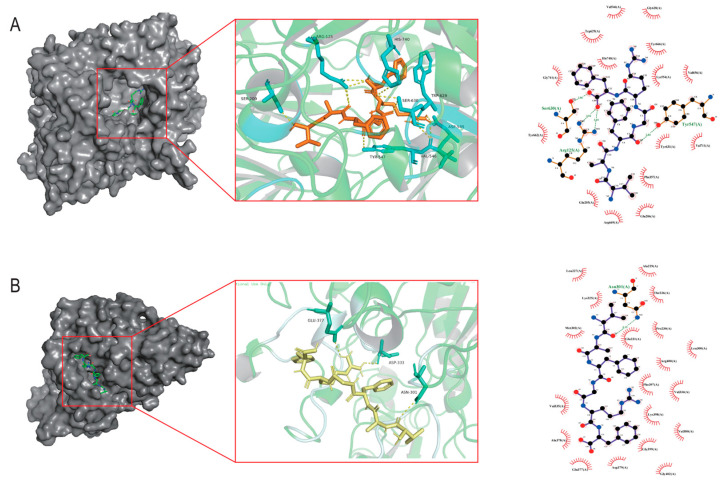
Molecular docking analysis of VF-6 with target protein receptors. (**A**) Three-dimensional binding conformation of VF-6 (orange sticks) to the active pocket of Dipeptidyl peptidase-IV (DPP-IV, PDB: 4a5s), with hydrogen bonds (yellow dashes) and hydrophobic interactions (red arc) labeled. (**B**) Interaction mode of VF-6 (yellow sticks) with α-glucosidase (PDB: 3wy1), highlighting catalytic residues Asn301, Asp333, and Glu377 within a distance of 3.5 Å.

**Figure 2 marinedrugs-23-00331-f002:**
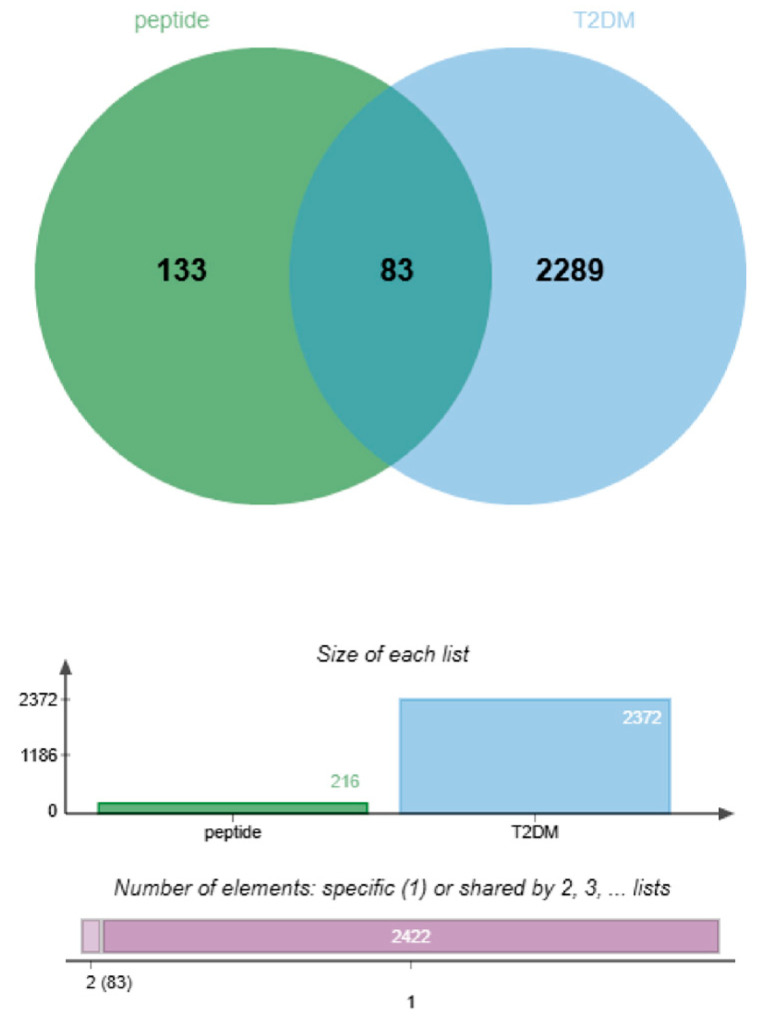
Identification of shared targets between bioactive peptides and T2DM. Venn diagram illustrating 83 overlapping targets derived from 216 peptide-predicted targets (green circle) and 2372 T2DM-associated targets (blue circle, sourced from DisGeNET and OMIM databases).

**Figure 3 marinedrugs-23-00331-f003:**
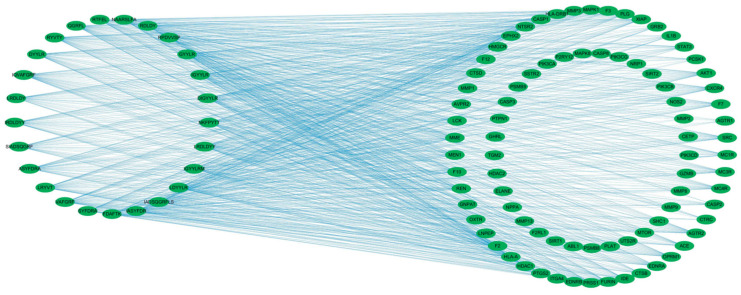
Component–target network of bioactive peptides and T2DM-related targets. The bipartite network comprises 25 candidate peptides (left nodes) and 83 common targets (right nodes), with edges representing peptide–target interactions. Network topology was analyzed using Cytoscape 3.9.1.

**Figure 4 marinedrugs-23-00331-f004:**
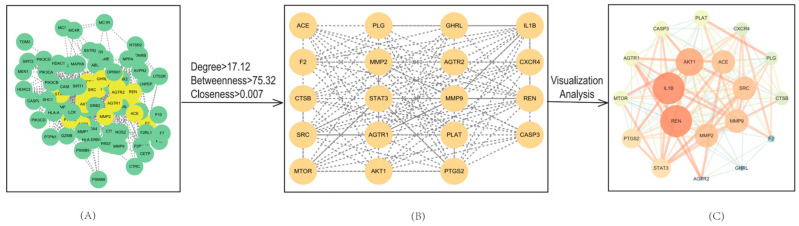
Protein–protein interaction (PPI) network of common targets. (**A**) Initial PPI network containing 82 nodes and 702 edges (STRING database; medium confidence). (**B**) Core subnetworks extracted via CentiScape (Degree = 17.12; Betweenness Centrality = 75.32; Closeness Centrality = 0.007). (**C**) Interaction map derived from visual analysis of key targets.

**Figure 5 marinedrugs-23-00331-f005:**
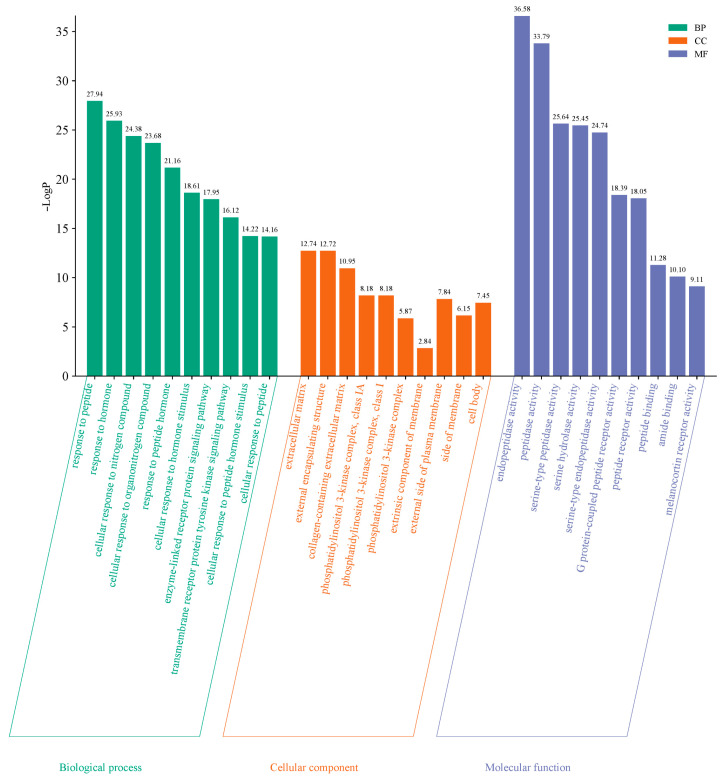
GO enrichment analysis of overlapping targets. The bar chart shows top 10 enriched terms in biological process (BP), molecular function (MF), and cellular component (CC) categories. Filter criteria: *p*-value < 0.01.

**Figure 6 marinedrugs-23-00331-f006:**
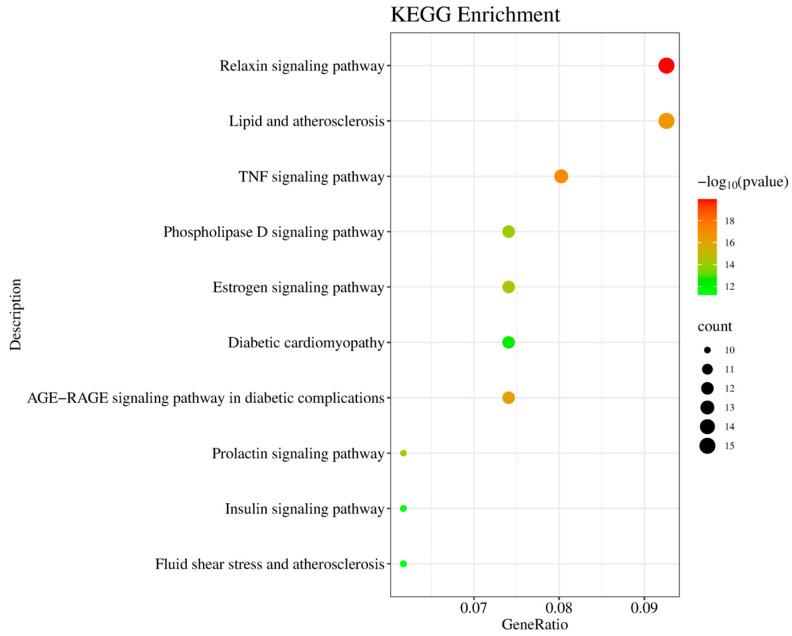
KEGG pathway enrichment analysis of common targets. Bubble graph displaying significantly enriched pathways (*p*-value < 0.01), including Relaxin signaling pathway (hsa04926), TNF signaling pathway (hsa04668), and insulin resistance (hsa04931).

**Figure 7 marinedrugs-23-00331-f007:**
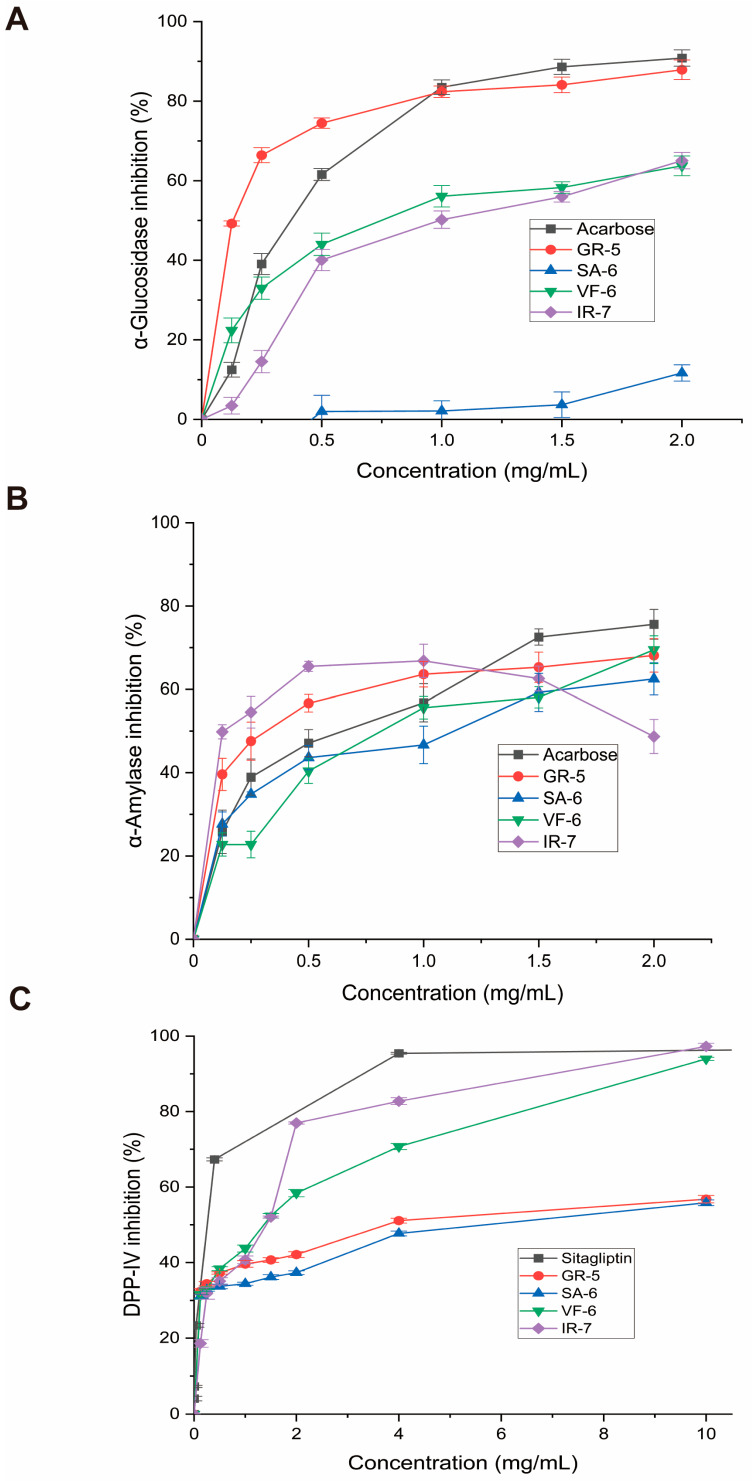
Enzymatic inhibition assays of four bioactive peptides. (**A**) α-Glucosidase inhibition rates (p-nitrophenyl-α-D-glucopyranoside substrate; 405 nm absorbance). (**B**) α-Amylase inhibition (starch-iodine method; 540 nm absorbance). (**C**) DPP-IV inhibition (fluorescence method; excitation wavelength is 360 nm; emission wavelength is 460 nm). Data expressed as mean ± SD (*n* = 3); acarbose and sitagliptin as positive controls.

**Figure 8 marinedrugs-23-00331-f008:**
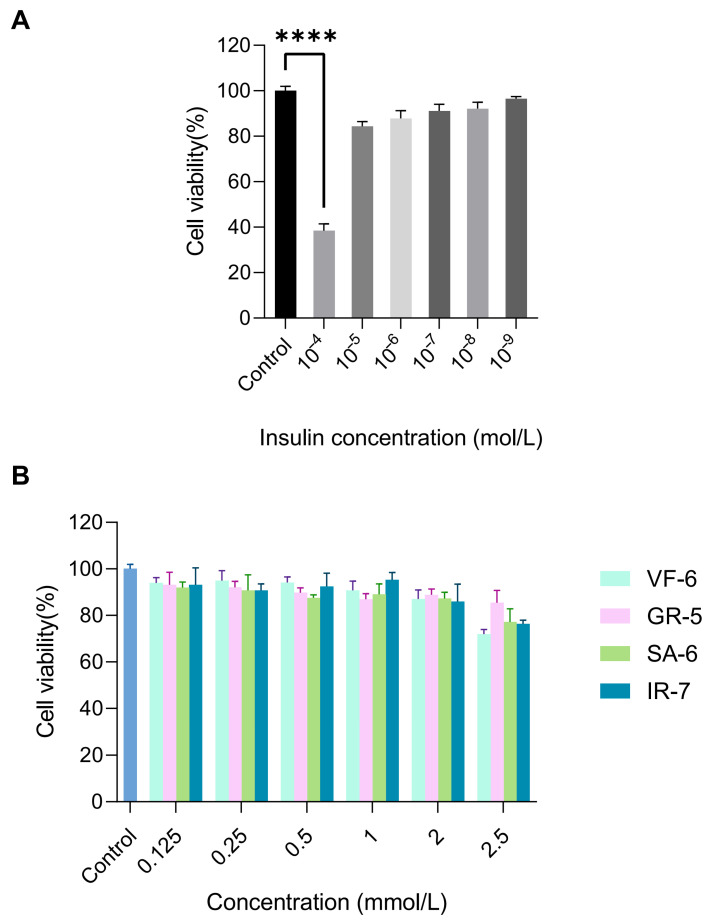
Cytotoxicity evaluation of test compounds in HepG2 cells via CCK-8 assay. (**A**) Cell viability under 10^−4^–10^−9^ mol/L insulin exposure, treatment with 10^−4^ mol/L insulin resulted in a marked reduction in HepG2 cell viability to 38.45%, demonstrating significant inhibition compared to the normal control (**** *p* < 0.0001). (**B**) Dose-dependent effects of peptides (0.125–2.5 mmol/L) on proliferation. Cells cultured in DMEM + 10% FBS, 5 × 10^3^ cells/well.

**Figure 9 marinedrugs-23-00331-f009:**
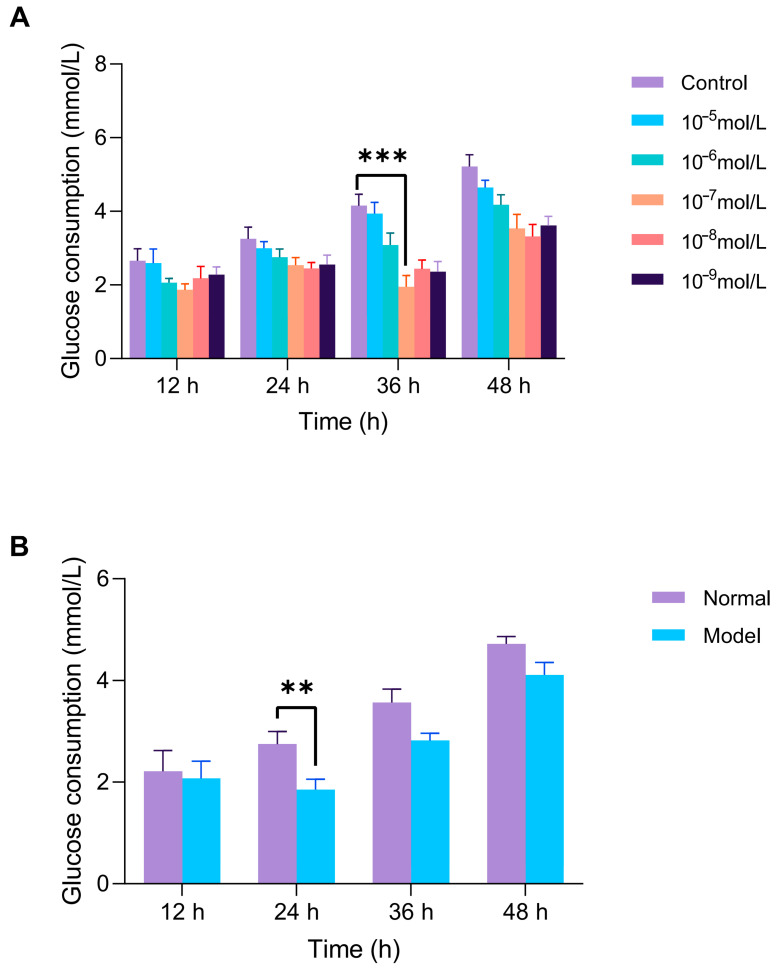
Establishment of insulin resistance (IR) model in HepG2 cells. (**A**) Optimization of insulin concentration (10^−5^–10^−9^ mol/L) and exposure time (12–48 h) via glucose uptake assay. (**B**) After optimal induction is complete, the stability of the model is verified by maintaining the cells in a complete medium (12–48 h). Glucose consumption is measured by glucose assay kits. ** *p* < 0.01, *** *p* < 0.001 vs. blank (normal HepG2) group.

**Figure 10 marinedrugs-23-00331-f010:**
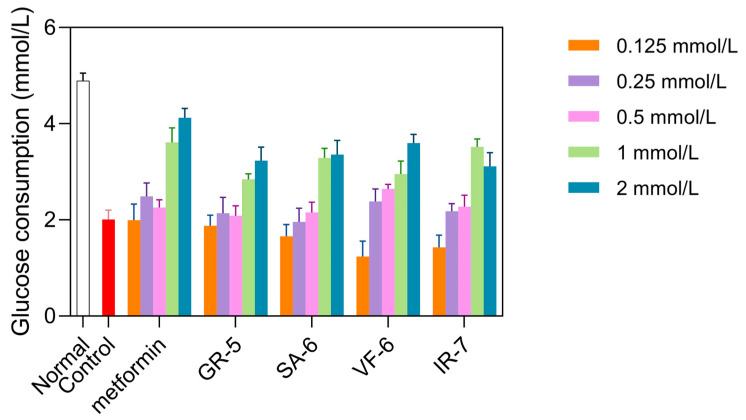
Glucose consumption in IR-HepG2 cells treated with peptides. Cells pretreated with 10^−7^ insulin for 36 h, followed by 24 h peptide incubation (0.125–2 mmol/L). Metformin as positive control.

**Figure 11 marinedrugs-23-00331-f011:**
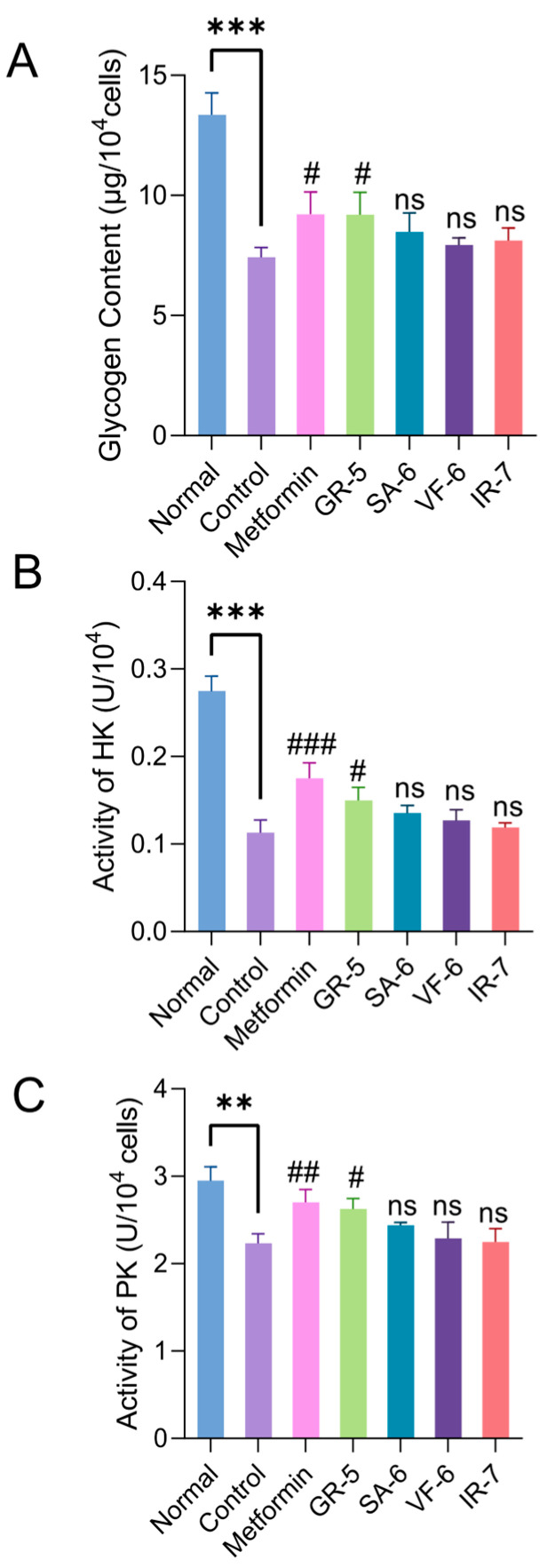
Regulatory effects of peptides on glucose metabolism in IR-HepG2 cells. (**A**) Intracellular glycogen content (anthrone-sulfuric acid method; 620 nm). (**B**) Hexokinase activity (NADH oxidation rate at 340 nm). (C) Pyruvate kinase activity (NADH oxidation rate at 340 nm). Data are presented as mean ± SD (n = 3). ** *p* < 0.01, *** *p* < 0.001 vs. blank (normal HepG2) group; # *p* < 0.05; ## *p* < 0.01; ### *p* < 0.001; *ns* not significant vs. IR-HepG2 control group.

**Table 1 marinedrugs-23-00331-t001:** Molecular docking results.

Peptide Sequence	Affinity		The Number of Hydrogen Bonds		The Shortest Hydrogen Bond Distance	
	3wy1	4a5s	3wy1	4a5s	3wy1	4a5s
VF-6	−8	−9.4	3	13	2.4 Å	2.2 Å
GR-5	−8	−8.9	8	8	2.2 Å	1.9 Å
SA-6	−8	−8.5	8	8	1.9 Å	2.4 Å
IR-7	−8.2	−8.9	7	8	2.1 Å	2.2 Å

**Table 2 marinedrugs-23-00331-t002:** Core target information.

No.	Target Name	Gene Name	Uniprot ID	Degree
1	Signal transducer and activator of transcription 3	STAT3	P40763	42
2	Caspase-3	CASP3	P42574	38
3	Serine/threonine-protein kinase mTOR	MTOR	P42345	32
4	Serine/threonine-protein kinase AKT	AKT1	P31749	56
5	C-X-C chemokine receptor type 4	CXCR4	P61073	30
6	Tyrosine-protein kinase SRC	SRC	P12931	45
7	Angiotensin-converting enzyme	ACE	P12821	33
8	Matrix metalloproteinase 2	MMP2	P08253	31
9	Tissue-type plasminogen activator	PLAT	P00750	25
10	Interleukin-1 beta	IL1B	P01584	54
11	Renin	REN	P00797	29
12	Cathepsin (B and K)	CTSB	P07858	27
13	Cyclooxygenase-2	PTGS2	P35354	32
14	Plasminogen	PLG	P00747	28
15	Thrombin	F2	P00734	23
16	Matrix metalloproteinase 9	MMP9	P14780	39
17	Angiotensin II receptor	AGTR2	P50052	18
18	Type-1 angiotensin II receptor	AGTR1	P30556	31
19	Appetite-regulating hormone	GHRL	Q9UBU3	19

## Data Availability

Data are contained within this article or Appendix A; further inquiries can be directed to the corresponding author.

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
