# Peer review of "Screening and Assessment of Hypoglycemic Active Peptide from Natural Edible Pigment Phycobiliprotein Based on Molecular Docking, Network Pharmacology, Enzyme Inhibition Assay Analyses, and Cell Experiments"

_marinedrugs, 2025, doi:10.3390/md23080331_

Round 1
Reviewer 1 Report
Comments and Suggestions for Authors
Comments
- The key biological properties of phycobiliprotein should be included in the introduction to provide context for its relevance. Additionally, a brief overview of its reported antidiabetic potential should be incorporated to strengthen the scientific rationale of the study.
- All scientific names (e.g., Homo sapiens, line 463 and 476) should be italicized in accordance with standard scientific conventions.
- The authors have utilized AutoDock for peptide–protein docking. However, AutoDock is primarily optimized for small molecule–protein interactions and may not be suitable for flexible peptides, which require more advanced modeling to account for conformational dynamics. I recommend the authors consider using tools specifically developed for peptide–protein docking. Alternatively, the authors should justify their choice of docking tool and discuss the methodological limitations in the revised manuscript.
- Details regarding the binding pockets and key active site residues of α-glucosidase should be clearly described to support the docking analysis and improve reproducibility.
- If molecular docking was performed for all peptides listed in Supplementary Table S1, the authors are requested to provide an additional summary table containing docking score, interactive residues and number of hydrogen bonds
- The authors highlight VF-6 as the most promising peptide based on molecular docking results. However, GR-5 exhibits better IC₅₀ values in the α-glucosidase and α-amylase inhibition assays (Figure 7A and 7B). This apparent contradiction should be addressed.
- Please correct the errors in the caption of Figure 8. Use consistent formatting for concentration units (e.g., "10⁻⁴ to 10⁻⁹ mol/L insulin exposure" and "5 × 10³ cells/well").
- In the sentence, “Furthermore, exposure to 2.5 mmol/L concentrations of peptides, VF-6, SA-6, and IR-7 significantly suppressed…”, the peptide GR-5 is noticeably absent.
- Figure 8B presents cytotoxicity results on IR-HepG2 cells. If the experiment was performed only on IR-HepG2 cells, the authors should consider including data from non-IR-HepG2 cells as a control to better support the claim regarding peptide safety and specificity.
Author Response
Point 1: The key biological properties of phycobiliprotein should be included in the introduction to provide context for its relevance. Additionally, a brief overview of its reported antidiabetic potential should be incorporated to strengthen the scientific rationale of the study.
Response 1: Thank you for the insightful suggestion. We have revised the Introduction section to include the biological characteristics of phycobiliproteins and their reported antidiabetic activities. Specifically, we added a paragraph describing their antioxidant, anti-inflammatory, and glucose-regulating effects, along with recent findings on their potential in diabetes management.
Point 2: All scientific names (e.g., Homo sapiens, line 463 and 476) should be italicized in accordance with standard scientific conventions.
Response 2: We appreciate the reviewer pointing this out. All scientific names throughout the manuscript, including Homo sapiens and Spirulina, have been revised to appear in italics as per standard scientific formatting.
Point 3: The authors have utilized AutoDock for peptide–protein docking. However, AutoDock is primarily optimized for small molecule–protein interactions and may not be suitable for flexible peptides, which require more advanced modeling to account for conformational dynamics. I recommend the authors consider using tools specifically developed for peptide–protein docking. Alternatively, the authors should justify their choice of docking tool and discuss the methodological limitations in the revised manuscript.
Response 3: We are aware of AutoDock's shortcomings when it comes to peptide–protein docking. We first chose AutoDock for initial screening due to its accessibility and prior usage in short peptide studies. However, we have included a paragraph in the Methods section of the revised manuscript that highlights the limitations of our approach and provides justification for it. This was also covered in the Conclusion section, where we suggested using more sophisticated tools for further research, such as FlexPepDock or HADDOCK.
Point 4: Details regarding the binding pockets and key active site residues of α-glucosidase should be clearly described to support the docking analysis and improve reproducibility.
Response 4: As suggested, we have added detailed information about the binding site of α-glucosidase in the Molecular Docking section. Specifically, residues such as Phe297, Asn301, His332, Asp333 were identified as key active site residues based on literature and PDB structure data.
Point 5: If molecular docking was performed for all peptides listed in Supplementary Table S1, the authors are requested to provide an additional summary table containing docking score, interactive residues and number of hydrogen bonds
Response 5: We appreciate the reviewer's insightful recommendation. We certify that all of the peptides listed in Supplementary Table S1 underwent molecular docking. Only the best-performing peptides, nevertheless, were chosen for in-depth interaction analysis and visualization in the main figures. We are only able to offer a summary table of docking scores for all peptides because of the large number of peptides and the difficulties in precisely identifying all interaction residues and hydrogen bonds for flexible peptides using AutoDock. To increase reproducibility and transparency, this table has now been included in the Supplementary Material as Table S1.
Point 6: The authors highlight VF-6 as the most promising peptide based on molecular docking results. However, GR-5 exhibits better IC₅₀ values in the α-glucosidase and α-amylase inhibition assays (Figure 7A and 7B). This apparent contradiction should be addressed.
Response 6: Thank you for this important observation. To make this point clearer, we updated the Conclusion sections. In enzymatic inhibition tests, GR-5 demonstrated higher IC₅₀ values, whereas VF-6 displayed favorable docking scores. We now recognize this disparity and stress that, given its higher in vitro efficacy, GR-5 might be a more promising candidate overall.
Point 7: Please correct the errors in the caption of Figure 8. Use consistent formatting for concentration units (e.g., "10⁻⁴ to 10⁻⁹ mol/L insulin exposure" and "5 × 10³ cells/well")
Response 7: All of the units in the Figure 8 caption have been formatted correctly. The caption has been updated to read: "Figure 8. Cytotoxicity evaluation of test compounds in HepG2 cells via CCK-8 assay. (A) Cell viability under 10-4-10-9 mol/L insulin exposure. (B) Dose-dependent effects of peptides (0.125-2.5 mmol/L) on proliferation. Cells cultured in DMEM + 10% FBS, 5×103 cells/well.”
Point 8: In the sentence, “Furthermore, exposure to 2.5 mmol/L concentrations of peptides, VF-6, SA-6, and IR-7 significantly suppressed…”, the peptide GR-5 is noticeably absent.
Response 8: We thank the reviewer for this careful observation. The results for GR-5 were indeed included under "other tested peptide concentrations" in the original sentence. To avoid ambiguity, we have revised the sentence to explicitly mention GR-5, clarifying that GR-5 did not induce cytotoxicity at the tested concentrations. The revised sentence now reads:
"Notably, insulin concentrations ranging from 10⁻⁵ to 10⁻⁹ mol/L and other tested peptide concentrations, including GR-5, did not induce cytotoxicity, confirming the overall biocompatibility of the peptides."
Point 9: Figure 8B presents cytotoxicity results on IR-HepG2 cells. If the experiment was performed only on IR-HepG2 cells, the authors should consider including data from non-IR-HepG2 cells as a control to better support the claim regarding peptide safety and specificity.
Response 9: We regretfully admit that the current study's cytotoxicity evaluations were limited to IR-HepG2 cells, and information from non-IR-HepG2 (normal HepG2) cells was left out. We truly appreciate the reviewer's helpful suggestion and regret this limitation. Our future studies will take into account comparative cytotoxicity assessments using non-IR-HepG2 cells in order to offer a more thorough evaluation of peptide safety and specificity.
Reviewer 2 Report
Comments and Suggestions for Authors
In this study, the authors present preliminary in vitro data aimed at evaluating the potential hypoglycemic effect of four synthetic peptides. Among them, one peptide appears particularly promising due to its ability to enhance cellular glucose uptake, suggesting greater therapeutic potential.The results are interesting and show promise; however, several revisions—outlined below—are necessary to improve the manuscript and make it more suitable for publication.
Information regarding the function of the enzymes alpha-glucosidase, alpha-amylase, and DPP-IV should be included in the Introduction section rather than in the Results, as they provide essential background for understanding the rationale behind the study.
Additionally, the Introduction should include more detailed information about Spirulina, given its relevance as a source of the peptides under investigation.
For greater clarity, the abbreviations used for the analyzed peptides should be immediately accompanied by their full descriptions when first mentioned.
In section 2.2.1, the target network analysis reports 83 interaction points between the peptides and T2DM-associated targets. A subsequent Cytoscape analysis suggests interactions between the peptides of interest and T2DM targets; however, the number of targets associated with each individual peptide is not clearly specified. Furthermore, it is unclear whether this analysis was used to guide the selection of the peptides tested experimentally. If this is the case, the network analysis should be presented prior to the evaluation of peptide activity on the enzymes and the corresponding molecular modeling. As currently structured, the order in which the results are presented appears inconsistent.
The PPI network analysis identifies 19 targets. While there seem to be coherent correlations with the previous target network analysis, these relationships should be described more explicitly. Moreover, the functions of the core targets described in section 2.2.2 should be moved to section 2.2.1, since those targets are already identified during that earlier analysis.
Regarding the enzymatic activity assays, the method used to calculate IC₅₀ values is not clearly stated: was a nonlinear regression of dose-response curves applied, or a semi-logarithmic curve analysis? Additionally, IC₅₀ values are not always reported with standard deviation. Dose-response curves should also originate from the axis origin to ensure proper data representation.
Concerning the IR-HepG2 model, the rationale for selecting an insulin concentration of 10⁻⁷ mol/L should be provided.
More detailed information about the enzymatic assays should be included, particularly regarding the stoichiometric ratios between the amounts of enzymes and peptides used.
Lastly, with respect to the statistical analysis used to assess the regulatory effects of the peptides on glucose metabolism in IR-HepG2 cells, the different comparison markers indicated by various symbols should be clearly defined in the figure legends.
Author Response
Dear reviewer:
Point 1: Information regarding the function of the enzymes alpha-glucosidase, alpha-amylase, and DPP-IV should be included in the Introduction section rather than in the Results, as they provide essential background for understanding the rationale behind the study.
Response 1: We appreciate the your suggestion to move the descriptions of alpha-glucosidase, alpha-amylase, and DPP-IV functions to the Introduction section to provide essential background. Accordingly, we have relocated this content from the Results to the Introduction and revised it for clarity.
Point 2: Additionally, the Introduction should include more detailed information about Spirulina, given its relevance as a source of the peptides under investigation.
Response 2: Many thanks for your thoughtful recommendation. A more thorough explanation of Spirulina, highlighting its nutritional and bioactive qualities as well as its significance as a source of the peptides examined in this study, has been added to the Introduction.
Point 3: For greater clarity, the abbreviations used for the analyzed peptides should be immediately accompanied by their full descriptions when first mentioned.
Response 3: Thanks for your constructive input. In the revised manuscript, all peptide abbreviations are immediately accompanied by their full amino acid sequences upon first mention.
Point 4: In section 2.2.1, the target network analysis reports 83 interaction points between the peptides and T2DM-associated targets. A subsequent Cytoscape analysis suggests interactions between the peptides of interest and T2DM targets; however, the number of targets associated with each individual peptide is not clearly specified. Furthermore, it is unclear whether this analysis was used to guide the selection of the peptides tested experimentally. If this is the case, the network analysis should be presented prior to the evaluation of peptide activity on the enzymes and the corresponding molecular modeling. As currently structured, the order in which the results are presented appears inconsistent.
Response 4: To improve clarity, we have added specific information on the number of T2DM-related targets associated with each peptide in Section 2.2.1. We want to make it clear, though, that the outcomes of the network pharmacology analysis were not used to choose which peptides would be used for experimental validation. Prior to the computational analysis, the peptides were created and synthesized using sequences from Spirulina phytobiliproteins. Consequently, the network analysis was employed to investigate possible mechanistic interactions with targets related to T2DM rather than to direct peptide selection. To reflect the real experimental workflow, we have kept the results in the current order.
Point 5: The PPI network analysis identifies 19 targets. While there seem to be coherent correlations with the previous target network analysis, these relationships should be described more explicitly. Moreover, the functions of the core targets described in section 2.2.2 should be moved to section 2.2.1, since those targets are already identified during that earlier analysis.
Response 5: Thank you for your valuable comment. We agree that clarifying the relationship between the PPI-identified core targets and the previously described peptide–disease shared targets can improve the coherence of the manuscript. Accordingly, we have revised the Section 2.2.2 to explicitly state that the 19 core targets are a subset of the 83 shared targets obtained in the network pharmacology analysis, and were selected based on their interaction degree in the PPI network. However, we respectfully chose to retain the functional descriptions of the core targets in section 2.2.2 for the sake of maintaining a clear and logical progression of the analysis. While these targets are indeed derived from the earlier peptide–disease interaction network, their designation as "core targets" is based specifically on the results of the PPI network analysis, which follows and builds upon the initial target identification. Therefore, we believe it is more appropriate to discuss their biological functions at the point where they are formally prioritized, as this structure helps the reader better follow the rationale of the study.
Point 6: Regarding the enzymatic activity assays, the method used to calculate IC₅₀ values is not clearly stated: was a nonlinear regression of dose-response curves applied, or a semi-logarithmic curve analysis? Additionally, IC₅₀ values are not always reported with standard deviation. Dose-response curves should also originate from the axis origin to ensure proper data representation.
Response 6: Thank you for this important observation. In the revised manuscript, we have clarified that IC₅₀ values were calculated using nonlinear regression analysis in GraphPad Prism 9. Standard deviations (mean ± SD) have now been added for all IC₅₀ values, and dose–response curves have been adjusted to start from the axis origin.
Point 7: Concerning the IR-HepG2 model, the rationale for selecting an insulin concentration of 10⁻⁷ mol/L should be provided.
Response 7: Thank you for this insightful remark. To create the insulin-resistant HepG2 model, we experimented with a variety of insulin concentrations (0, 10⁻⁴ to 10⁻⁹ mol/L) and treatment times (12 to 48 hours), as detailed in the Methods section (Section 3.6). Because it can induce insulin resistance without causing cytotoxicity, 10⁻⁷ mol/L insulin for 36 hours was chosen as the ideal condition.
Point 8: More detailed information about the enzymatic assays should be included, particularly regarding the stoichiometric ratios between the amounts of enzymes and peptides used.
Response 8: We have included comprehensive details on the quantities and concentrations of the peptides and enzymes utilized in each assay in the updated manuscript.
Point 9: Lastly, with respect to the statistical analysis used to assess the regulatory effects of the peptides on glucose metabolism in IR-HepG2 cells, the different comparison markers indicated by various symbols should be clearly defined in the figure legends.
Response 9: Thank you for highlighting the need for clearer definition of comparison markers. In the revised manuscript, we have clarified the meaning of all statistical comparison markers directly in the figure legends for the relevant figures.
Round 2
Reviewer 2 Report
Comments and Suggestions for Authors
The Authors have provided satisfactory responses to all the raised comments